# Cytotoxicity Induced by Black Phosphorus Nanosheets in Vascular Endothelial Cells via Oxidative Stress and Apoptosis Activation

**DOI:** 10.3390/jfb14050284

**Published:** 2023-05-20

**Authors:** Hao Dong, Yin Wen, Jiating Lin, Xianxian Zhuang, Ruoting Xian, Ping Li, Shaobing Li

**Affiliations:** 1Center of Oral Implantology, Stomatological Hospital, School of Stomatology, Southern Medical University, Guangzhou 510280, China; donghaoct3@163.com (H.D.); wennyinn@gmail.com (Y.W.);; 2First Clinical Medical College, Xinjiang Medical University, Urumqi 830011, China; 3The First People’s Hospital of Kashgar Region, Kashgar 844000, China

**Keywords:** nanomaterials, black phosphorus, cytotoxicity, vascular endothelial cell, reactive oxygen species, apoptosis, biomedical applications

## Abstract

Black phosphorus (BP), an emerging two-dimensional material with unique optical, thermoelectric, and mechanical properties, has been proposed as bioactive material for tissue engineering. However, its toxic effects on physiological systems remain obscure. The present study investigated the cytotoxicity of BP to vascular endothelial cells. BP nanosheets (BPNSs) with a diameter of 230 nm were fabricated via a classical liquid-phase exfoliation method. Human umbilical vein endothelial cells (HUVECs) were used to determine the cytotoxicity induced by BPNSs (0.31–80 μg/mL). When the concentrations were over 2.5 μg/mL, BPNSs adversely affected the cytoskeleton and cell migration. Furthermore, BPNSs caused mitochondrial dysfunction and generated excessive intercellular reactive oxygen species (ROS) at tested concentrations after 24 h. BPNSs could influence the expression of apoptosis-related genes, including the P53 and BCL-2 family, resulting in the apoptosis of HUVECs. Therefore, the viability and function of HUVECs were adversely influenced by the concentration of BPNSs over 2.5 μg/mL. These findings provide significant information for the potential applications of BP in tissue engineering.

## 1. Introduction

Maxillofacial bone defects are a common pathology that often leads to tooth loss, masticatory difficulties, and even facial deformities [1]. Current clinical treatments, such as bone grafting or bone transplantation, are associated with postoperative complications. Therefore, the safer and more effective therapeutic approaches are required [2]. Black phosphorus (BP) as a novel two-dimensional nanomaterial produces phosphate ions upon biodegradation, which serve as substrates for osteogenesis and facilitate in situ treatment of bone defects [3,4,5]. This approach provides a new strategy for the treatment of bone tissue diseases. BP has a similar lamellar structure to graphene [6,7]; its regular ribbed surface shape makes BP an excellent vector for drug delivery [8,9] and gene editing [10]. Scaffolds doped with BP nanoparticles have a higher mechanical strength [11,12]. Based on its excellent photonic properties, BP has been investigated for cancer treatment using photothermal [13,14] and photodynamic [15] therapies. Therefore, a multifunctional disease-therapeutic nanoplatform based on BP could be developed in the treatment of maxillofacial bone defects [16].

Phosphorus is a widely present element in the human body. It participates in almost all physiological and chemical reactions and plays a vital role in maintaining the physiological functions of the human body [16]. It is generally believed that the degradation of BP in vivo produces non-toxic phosphate [17,18]. However, some previous studies did not support this conclusion. BP nanosheets (BPNSs) showed obvious cytotoxicity to L929 fibroblasts at concentrations above 4 μg/mL [19], and BP quantum dots (BPQDs) significantly inhibited the growth of Beas-2E cells at a concentration of 5 μg/mL [20]. A study suggests that the toxicity of BP to organisms is related to its concentration, size [21], and surface modification [22]. Therefore, a better understanding of the biotoxicity of BP is required to facilitate the tissue regeneration. After nanomaterials enter the body, they pass through the blood circulation to reach the target [13]. Vascular endothelial cells constitute the inner wall of the blood vessels. Previous studies have shown that nanoparticles can be transferred into the blood [23] so the BP entering the body inevitably comes into contact with vascular endothelial cells. At the same time, the process of osteogenesis relies heavily on the infiltration of early-stage vascular endothelial cells, which provide nutrients to local tissues and remove metabolic waste through neovascularization [24]. However, there are few studies on the toxicity of BP during the process of angiogenesis.

In this study, the effects of BPNSs on the metabolic activity, cell migration, and cell apoptosis and its related gene expression of human umbilical vein endothelial cells (HUVECs) were investigated. The BPNSs were prepared using the classic liquid-phase exfoliation method. HUVECs were used to simulate the vascular structure in vitro [25]. The first objective of this study was to determine the cytotoxic effects of BPNSs on vascular endothelial cells. The second objective was to investigate the intracellular response and the underlying mechanisms of BPNSs-induced cytotoxicity.

## 2. Materials and Methods

### 2.1. The Synthesis of BPNSs

BPNSs were fabricated at a low temperature using a classical liquid-phase exfoliation method. BP crystals (50 mg) (99.998%, XFNANO, Nanjing, China) were mixed with 100 mL of N-methyl-2-pyrrolidone (NMP, Electronic grade 99.9%, Aladdin, Shanghai, China). The mixture was treated using an ultrasonic cell disruptor (JY92-IIDN, Xinzhi, Ningbo, China) in an ice water mixture for 15 h (21.0–22.5 kHz, working for 5 s and paused for 10 s). After a brown suspension was formed, unstripped BP crystals were removed by centrifugation at 4500 rpm for 30 min at 4 °C, and the resulting supernatant was collected. Next, the NMP was eliminated through centrifugation at 15,000 rpm for 30 min at 4 °C. The BPNSs obtained were then subjected to three rounds of centrifugation using absolute ethanol and deionized ultrapure water in sequence. After quantification of the manufactured BPNS powder by lyophilization, it was diluted in the culture medium to different concentrations (0.31, 0.63, 1.25, 2.5, 5, 10, 20, 40 and 80 μg/mL).

### 2.2. Characterizations of BPNSs

Multiple microscopy techniques were used to assess the surface morphology and thickness of BPNSs. Atomic force microscopy (AFM) was employed to perform a vacuum-based characterization of the surface morphology and thickness of the BPNSs at room temperature, utilizing the Dimension Icon system (Bruker, Karlsruhe, Germany). Meanwhile, scanning electron microscopy (SEM) was used to examine the surface morphology of the BPNSs by depositing them onto an aluminum foil and drying at 60 °C prior to imaging. The imaging was carried out under high vacuum with an acceleration voltage of 10 kV using the Sigma 300 system (Zeiss, Oberkochen, Germany). Lastly, transmission electron microscopy (TEM) was employed to determine the elemental compositions and morphologies of the BPNSs utilizing the FEI Tecnai F20 TEM D545 system (FEI, Hillsboro, OR, USA). The Raman spectra of the BPNSs were obtained at room temperature using Raman spectroscopy (LabRAM HR, HORIBA, Montpellier, France) with an excitation wavelength of 532 nm. The particle size distribution and polydispersity (PDI) in water were determined using a Zetasizer 3000 HS nanosizer (Malvern Instruments, Malvern, UK).

### 2.3. Cell Culture

The human umbilical vein endothelial cells (HUVECs, National Infrastructure of Cell Line Resource, Beijing, China) were cultured in Dulbecco’s modified Eagle’s medium (DMEM, Gibco, Waltham, MA, USA), supplemented with 10% (*v*/*v*) heat-inactivated fetal bovine serum (FBS, Gibco, Waltham, MA, USA) and antibiotics (100 IU/mL penicillin and 100 μg/mL streptomycin) (P/S, Gibco, Waltham, MA, USA) under standard conditions of 37 °C and 5% CO_2_ in a humidified atmosphere.

### 2.4. Cell Uptake

Cellular uptake assays were performed to verify the intracellular entry of the BPNSs. BPNSs were labeled with fluorescein isothiocyanate (FITC, Thermo Fisher, Waltham, MA, USA), and FITC-labelled BPNSs (10 μg/mL) were exposed to the cells for 0.5, 1, 2 and 4 h. The treated cells were stained with 4′,6-diamidino-2-phenylindole dihydrochloride (DAPI, SolarBio Technologies, Beijing, China) and images were captured under a laser confocal microscope (STELLARIS 5, Leica, Weztlar, Germany). The relative uptake levels were calculated as the fluorescence intensity. Five fields of views were captured per well and fluorescence images were analyzed using the ImageJ 1.46 software.

### 2.5. Cell Morphology and Metabolic Activity

Morphological changes in HUVECs were observed to evaluate the cytotoxicity of BPNSs. HUVECs were treated with various concentrations of BPNSs (ranging from 0 to 80 μg/mL) in the cell culture medium for 24 h, and images were captured using an inverted microscope (DMIL LED, Leica, Weztlar, Germany). The metabolic activity of HUVECs was measured by using the CCK-8 assay (Dojindo, Kumamoto, Japan). First, 1 × 10^4^ HUVECs were seeded in a 96-well plate and incubated for 24 h, and five wells were set in each experimental group for auxiliary purposes. Subsequently, the cells were treated to a culture medium with varying concentrations of BPNSs for 24 h. After the cell culture period, the cells were incubated with the CCK-8 solution at 37 °C for 1 h in the dark. Absorbance at 450 nm was measured using a microplate reader (iMark, Bio-Rad, Hercules, CA, USA). To calculate cell metabolic activity, the absorbance values of the cells treated with BPNSs were expressed as a percentage relative to the negative control groups (without BPNSs), as previously reported [26].

### 2.6. Cell Migration Ability

The cell migration ability was detected using a scratch test to reflect the cytotoxicity of BPNSs. According to a classic protocol, plate scratches were created using sterile blades at the bottom of 6-well plates before the cells were seeded. HUVECs were seeded in plates at a density of 5 × 10^5^ cells/well and incubated for 24 h. Then, cell scratches perpendicular to the plate scratches were made using a 1 mL pipette tip. BPNSs were diluted to different concentrations and added to the cells for 24 h. Images of the intersection point of the plate scratch and cell scratch in each well were taken at 0 and 24 h. The area of cell migration was calculated as the difference between two time points.

### 2.7. Cytoskeleton Staining

Morphological changes in the cytoskeleton of HUVECs were observed by phalloidin staining (Abcam, Cambridge, UK). HUVECs (5 × 10^4^ cells/well) were seeded into 24-well plates lined with climbing slides and incubated for 24 h. Different concentrations of BPNSs (1.25, 5 and 20 μg/mL) were added to the cells and incubated for 24 h. The treated HUVECs were stained with phalloidin-iFluor 488 (Abcam, Cambridge, UK) and DAPI for microscopic fluorescence images.

### 2.8. Reactive Oxygen Species Test

To determine the intracellular reactive oxygen species (ROS) levels in HUVECs, a ROS assay kit (Meilunbio, Dalian, China) was utilized following the manufacturer’s guidelines. Specifically, HUVECs were seeded into 12-well plates at a density of 1 × 10^5^ cells per well and incubated for 24 h prior to the addition of BPNSs. The cells were then treated with BPNSs for another 24 h before analysis. Next, the previous culture medium was replaced with a medium containing 1% of the fluorescent probe (2,7-dichlorodihydrofluorescein diacetate, DCFH-DA). After incubation for 1 h at 37 °C in the dark, the cells were washed with a culture medium three times. Fluorescence images were captured immediately, utilizing the DMi8 inverted fluorescence microscope under FITC filters (Leica, Weztlar, Germany). The relative ROS levels were calculated as the fluorescence fold relative to the control group. Five fields of views were captured per well and fluorescence images were analyzed using the ImageJ 1.46 software.

### 2.9. Mitochondrial Membrane Potential Detection

Mitochondrial membrane potential (ΔΨm) was detected using the JC-1 assay kit (Meilunbio, Dalian, China). HUVECs were placed in 12-well plates (1 × 10^5^ cells/well) and incubated for 24 h. Afterward, the initial medium was removed, and replaced with a fresh medium containing various concentrations of BPNSs for an additional 24 h. Cells were washed twice with PBS and then stained with a medium containing 0.5% of JC-1 for 30 min at 37 °C in the dark. Cells were washed twice in a stain buffer and covered with the culture medium. Finally, the cells were immediately observed under a confocal microscope. Images were taken at 490 nm and 525 nm excitation wavelengths, and 530 nm and 590 nm emission wavelengths. Changes in mitochondrial membrane potential were expressed as the ratio of red-to-green fluorescence intensity. Five fields of views were captured per well and fluorescence images were analyzed using the ImageJ 1.46 software.

### 2.10. Cell Apoptosis Test

Cell apoptosis rate was detected using an apoptosis assay kit (Meilunbio, Dalian, China). According to the manufacturer’s instructions, HUVECs were placed in 12-well plates (1 × 10^5^ cells/well) and incubated for 24 h. Different concentrations of BPNSs (1.25, 5 and 20 μg/mL) were added to the cells and incubated for 24 h. The treated HUVECs were collected and then washed twice with PBS. Cells were stained with a medium containing 1% of the Annexin-V/PI dye for 15 min at 37 °C in the dark and the apoptosis rate was analyzed using flow cytometry (DxFLEX, Beckman, Brea, CA, USA).

### 2.11. Apoptosis-Related Gene Expression

The reverse transcription-quantitative polymerase chain reaction (RT-qPCR) assay, in accordance with the manufacturer’s instructions, was employed to evaluate the expression levels of apoptosis-related genes in HUVECs. HUVECs were cultured in 12-well plates at a density of 1 × 10^5^ cells per well and incubated for 24 h. Different concentrations of BPNSs (1.25, 5 and 20 μg/mL) were added to the cells and incubated for 24 h. The total RNA was extracted from the cells using the Trizol reagent (Accurate Biology, Changsha, China) and converted to cDNA using the Reverse Transcription Mix Kit (Accurate Biology, Changsha, China). Real-time PCR was performed on a LightCycler 96 real-time PCR system (Roche, Basel, Switzerland) using the SYBR Green Premix Kit (Accurate Biology, Changsha, China). The 2^−ΔΔCT^ method, with 18S ribosomal RNA (18s rRNA) as the reference gene, was employed to calculate the relative changes in mRNA expression. Three experimental wells were set up for each group. Control groups did not receive BPNS pretreatment. The primers used in this study are listed in Table 1 and were purchased from Sangon Biotech (Shanghai, China).

### 2.12. Statistical Analysis

The statistical analyses were conducted using GraphPad 9.0 (GraphPad Software, San Diego, CA, USA), and the data are presented as mean ± standard deviation. The significance of the results was determined using one-way ANOVA followed by Tukey’s multiple comparison test, with a threshold for statistical significance set at *p* < 0.05. Each cell experiment was independently conducted in triplicate for statistical reliability.

## 3. Results

### 3.1. BPNSs Characterization

A photograph of the as-exfoliated BPNS dispersion is shown in Figure 1a, where the BPNSs dispersion is dark brown and shows good homogeneity and stability. The surface morphology of layered BP is shown in Figure 1b,d. The size of the lateral particles is distributed from 100 to 500 nm. The height profiles in Figure 1c were obtained from the AFM image in Figure 1b, showing that the thickness of BPNSs was from ~3 to 6 nm (5–10 layers) [27,28]. The TEM image in Figure 1e shows a similar diameter as that in the AFM and SEM images. The elemental distributions in Figure 1e were detected, and the main elements in the particle area were phosphorus and oxygen. The Raman spectrum of the BPNSs is shown in Figure 1f. The three prominent peaks can be ascribed to the $A_{g}^{1}$ out-of-plane phonon modes at 359 cm^−1^, $B_{2g}$ in-plane modes at 435 cm^−1^, and $A_{g}^{2}$ in-plane modes at 463 cm^−1^. These peaks are generally considered the unique high-frequency interlayer Raman modes of BP [29]. The particle size distributions of the BPNSs are shown in Figure 1g with a mean size of 239.09 nm. The polydispersity (PDI) of BPNSs was measured as 0.123, indicating excellent dispersion [21].

### 3.2. Cell Uptake

FITC-labeled BPNSs were constructed and FITC fluorescence was observed in the cytoplasm for different time points, as shown in Figure 2a. The intracellular fluorescence intensity enhanced with an increase in the exposure time. The quantitative analysis also proved this finding (Figure 2b), indicating the accumulation of BPNSs in the cells.

### 3.3. Cell Morphology and Metabolic Activity

To investigate the cell morphology of BPNSs, HUVECs were treated with various concentrations of BPNSs for 24 h. As shown in Figure 3, cells without BPNSs had a blunt and cobblestone-like morphology. The number and morphology of HUVECs were not significantly different when the BPNSs concentration was less than 2.5 μg/mL. The number of black particles and aggregated BPs gradually increased with increasing concentrations. When the concentrations were greater than 10 µg/mL, black particles were observed in the cellular region, indicating the cell uptake of BPNSs.

HUVECs were exposed to increasing BPNSs concentrations (0.3–80 µg/mL) for 24 h, and the cellular metabolic activity was evaluated. Figure 4 shows that relative cell metabolic activity decreases with increasing concentrations at 24 h. This result indicated that the cytotoxicity of BP was concentration-dependent. Compared to the control group, the cell metabolic activity was higher than 80% at concentrations lower than 5 µg/mL, while a 50% reduction in cell metabolic activity was observed at a concentration of 40 µg/mL at 24 h.

### 3.4. Cell Migration Ability

The migration of vascular cells is essential for their proper function. Figure 5a shows the cell scratches at different BPNSs concentrations. At 24 h after scratch formation, the scratch area healed at all concentrations. The migration area of the cells was not significantly different from that of the control group until 1.25 μg/mL (Figure 5b). A significant decrease in migration area was detected at higher concentrations, indicating that BPNSs inhibited the migration of cells. After BPNSs were added to the culture medium for 24 h, the number of dead cells (round and bright in morphology [30]) increased with the concentration.

### 3.5. Cytoskeleton Staining

The cytoskeleton is an intracellular structure that maintains a balance between centrifugal and centripetal forces within the cell. The cytoskeleton can stabilize the cell membrane by regulating junctional complexes. As shown in Figure 6, more radial actin fibers were found in the control group cells. The cells maintained a similar state when 1.25 μg/mL of BPNSs were introduced. However, disordered fibers emerged at 5 μg/mL. At 20 μg/mL, the cells retained their morphology but had vague spot-like F-actin filaments.

### 3.6. ROS and Mitochondrial Membrane Potential Test

Once inside the cell, the fluorescent probe DCFH-DA was hydrolyzed to DCFH. In the presence of ROS, DCFH was oxidized to generate DCF, which emits green fluorescence. Therefore, the intensity of green fluorescence derived from DCF can be used to gauge the intracellular ROS level. As shown in Figure 7a, green fluorescence in the cells was not obvious at BP concentrations of less than 2.5 μg/mL. However, with the increase in BP concentration, the green fluorescence intensity gradually increased. The highest concentration (80 μg/mL) showed the most vigorous fluorescence intensity, which was 10.5 times higher than that of the control group (Figure 7b).

The alteration in mitochondrial membrane potential of cells following 24 h of treatment with BPNSs is illustrated in Figure 7d. Red fluorescence was dominant at concentrations below 1.25 μg/mL; red and green fluorescence was observed within the concentration range of 2.5–20 μg/mL, and green fluorescence was prevalent at 40 and 80 μg/mL. In Figure 7c, the mitochondrial membrane potential was reduced for the groups with BPNSs, and there was a statistical difference between the group with the lowest BPNSs concentration and the control group.

### 3.7. Cell Apoptosis Test

Elevated levels of ROS and mitochondrial dysfunction are both implicated in the induction of cell apoptosis. Flow cytometry determines the cell state by detecting the fluorescence of cells, and apoptotic cells appear in the two quadrants on the right side. The results of flow cytometry analysis showed an increased number of the experimental group cells in the two quadrants on the right side (Figure 8a), indicative of a greater presence of apoptotic cells. Quantitative analysis further demonstrated a statistically significant increase in the proportion of apoptotic cells within the experimental group compared to the control group (Figure 8b), with significant differences observed at concentrations of 5 and 20 μg/mL.

### 3.8. Apoptosis-Related Gene Expression

The objective of this study was to evaluate the impact of BPNSs on the expression of apoptosis-associated genes in the HUVEC cells through the RT-PCR analysis, as shown in Figure 9. After 24 h of exposure with BPNSs of varying concentrations, the experimental groups exhibited an incremental rise in the expression of the P53 gene compared to the control group. Additionally, concentrations of 5 and 20 μg/mL showed statistically significant differences from the control group. In contrast, the expression of the BCL-2 gene was significantly higher in the control group than in the experimental group, with a statistically significant difference observed at concentrations of 5 and 20 μg/mL. The pattern of BAX expression followed that of P53, with higher expression levels in the experimental group than the control group, resulting in statistically significant differences at concentrations of 5 and 20 μg/mL.

## 4. Discussion

The unique characteristic of a single-element composition and exceptional optoelectronic properties render two-dimensional BP nanomaterials a promising candidate for deployment in craniofacial applications [31]. Biosafety is a key parameter that determines the application of nanomaterials in vivo. Nevertheless, the range of biocompatibility of BP nanomaterials remains unclear [32]. Hence, the current study investigated the cytotoxicity and the underlying mechanisms of black phosphorus nanosheets on vascular endothelial cells. In this study, BPNSs with stable dimensions and properties were obtained via liquid phase exfoliation. Concurrently, the results revealed that exposure of HUVECs with BPNSs at concentrations exceeding 2.5 μg/mL led to a significant reduction in cellular metabolic activity. Our findings indicated that higher concentrations of BPNSs increased the levels of oxidative stress and apoptosis within HUVECs. Additionally, RT-PCR analyses demonstrated that BPNSs had a significant effect on the expression of genes associated with apoptosis.

BPNSs were fabricated using the classical liquid-phase stripping method. Morphological characterization showed that the diameter range of the BPNSs was 100–500 nm, and the height was 3–6 nm, corresponding to 5–10 layers. The BPNSs exhibited good stability and dispersibility, as characterized by Raman spectroscopy and particle-size measurements in Figure 1. The BPNSs in this study had size and properties similar to those of BP nanomaterials in other biomedical applications [5,11].

Nanomaterials can directly enter the cells or interact with them through extracellular contact [33]. To demonstrate the effects of BPNSs on HUVECs, FITC-labelled BPNSs were constructed and the green fluorescence of FITC was observed in the cytoplasm as shown in Figure 2a, implying that BPNSs were internalized by cells. A previous report suggested that the ribbed surface of BP could reduce damage to the cell membrane [34], while the opposite view revealed that BPNSs could damage the integrity of the cell membrane when entering cells [20]. Furthermore, damage to the cell membrane affected the cytoskeleton [35,36], damaging cell morphology and motor function. Disordered myofilaments appeared in the cells when the concentration was higher than 5 μg/mL. Cytoskeletal alterations, as well as excessive intake of BPNSs, could affect cell motility [37]. When the concentration of BPNSs exceeded 2.5 μg/mL, the scratch assay revealed a significant inhibition of cell migration ability (Figure 5b), which could undermine the suitability of BPNSs for tissue engineering applications. Cell viability continued to decrease with an increasing incubation time, further reducing migration ability. Therefore, under the abovementioned actions, BPNSs ultimately decreased cell activity and caused cytotoxicity.

The ROS induced by nanomaterials plays an important role in cytotoxicity [38]. An appropriate level of ROS may be crucial for the physiological functions of biological components [39]. Although the body can balance oxidative stress by processing redox products, excessive ROS can cause various adverse effects [40]. Our results showed a significant increase in intracellular ROS levels after 24 h of exposure to BPNSs at all concentrations in Figure 7b. A negative correlation between ROS levels and cell viability was demonstrated (Figure 10), consistent with the other toxicity studies [41]. Based on previous research, excessive production of ROS has been implicated in the activation of the apoptotic pathway, ultimately resulting in cell death [42], which was also observed in our cell viability and migration ability tests. Mitochondria are the main sites of cellular ROS production. Excessive ROS causes mitochondrial destruction, and mitochondrial fission accelerates ROS accumulation. Such a vicious cycle could amplify the effects of oxidative stress and eventually lead to mitochondrial dysfunction [43]. The results showed that even the lowest concentration of BPNSs (0.31 μg/mL) could decrease mitochondrial membrane potential (Figure 7c), and this effect was more obvious at a higher concentration (up to 80 μg/mL).

Excessive ROS generation exerts direct damage on intracellular biomolecules, resulting in adverse effects on cellular physiological function and the induction of apoptosis [43]. In addition, mitochondrial dysfunction adversely affects intracellular energy metabolism, contributing to the onset of apoptosis [32]. The P53 gene is known to play a pivotal role in regulating cell fate. In response to DNA damage, it can elicit various responses including autophagy, senescence, and apoptosis [44]. Furthermore, P53 has been shown to be causally associated with G1 arrest, which in turn affects embryonic development [45]. Additionally, P53 has been implicated in promoting apoptosis via the regulation of ferroptosis [46]. In response to oxidative stress, the P53 gene-encoded protein could influence the process of apoptosis by mediating mitochondrial membrane potential and activating BAX [47]. Both BCL-2 and BAX are constituents of the BCL-2 gene family, contributing to the intricate regulation of cell apoptosis. In response to apoptotic signals, members of the BCL-2 family become activated and mediate the permeabilization of the outer mitochondrial membrane, a critical step in the process of apoptosis [48]. The protein encoded by the BCL-2 gene restrains apoptosis, whereas the protein encoded by the BAX gene promotes it. These proteins hold opposing effects on each other, affecting the course of apoptosis [49]. The flow cytometry analysis demonstrated a significant increase in apoptosis when the concentration of BPNSs exceeded 5 μg/mL in Figure 8b. Additionally, the RT-PCR results revealed that BPNSs induced the up-regulation of P53 gene expression in cells. Moreover, at concentrations above 5 μg/mL, statistically significant down-regulation of BCL-2 and up-regulation of BAX were observed relative to the control group. As an upstream gene of BCL-2 gene family, the up-regulation of P53 expression promotes cell apoptosis. Therefore, the down-regulation of BCL-2, which inhibits cell apoptosis, leads to the up-regulation of BAX expression that antagonizes BCL-2, indicating that BPNSs promote cell apoptosis. The expression levels of apoptosis-related genes also reflected that BPNSs promoted apoptosis in a concentration-dependent manner. Hence, the cytotoxicity of BPNSs on HUVECs is contingent on several factors, including the integrity of the cell membrane, intracellular oxidative stress levels, and the down-regulation of mitochondrial membrane potential. These factors collectively alter the cellular activity and function by inducing apoptosis (Figure 11).

Nevertheless, the size and surface properties of nanomaterials could additionally influence their cytotoxicity. Apart from inducing apoptosis, they could also generate cytotoxicity by affecting the cell cycle and triggering intracellular inflammatory responses. Therefore, more investigations are warranted to gain a comprehensive understanding of the toxicity and to provide a framework for future research based on BP.

## 5. Conclusions

In this study, the cytotoxicity of BPNSs to HUVECs and its underlying mechanisms were investigated, and the following conclusions were drawn:BPNSs exhibited significant cytotoxicity towards HUVECs at concentrations exceeding 2.5 μg/mL, characterized by inhibited cell metabolic activity, disrupted cytoskeleton, and suppressed cell migration.The cytotoxicity mechanism of BPNSs on HUVECs involves the generation of excessive ROS and mitochondrial dysfunction, ultimately leading to apoptosis.The cytotoxic effect of BPNSs on HUVECs was associated with apoptosis-associated genes, including P53 and the BCL-2 family.

## Figures and Tables

**Figure 1 jfb-14-00284-f001:**
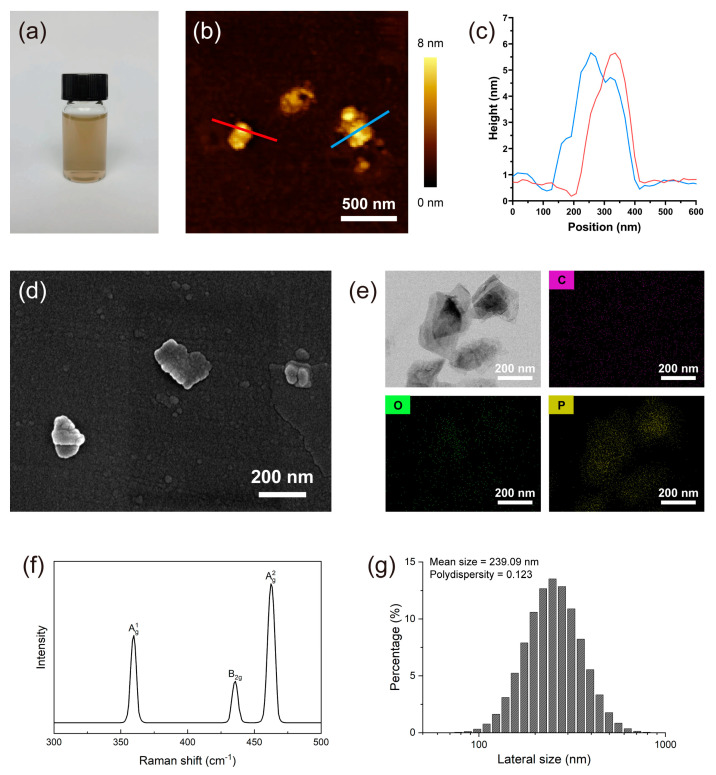
Physicochemical characterizations of the BPNSs. (**a**) Macroscopic image of 10 μg/mL BPNSs dispersed in water; (**b**) AFM image of BPNSs and (**c**) height profiles of BPNSs obtained from the AFM image; (**d**) SEM image of BPNS morphology; (**e**) TEM image and surface elements mapping images of BPNSs; (**f**) Raman spectrum of BPNSs; (**g**) Lateral size distributions and polydispersity of BPNSs in water.

**Figure 2 jfb-14-00284-f002:**
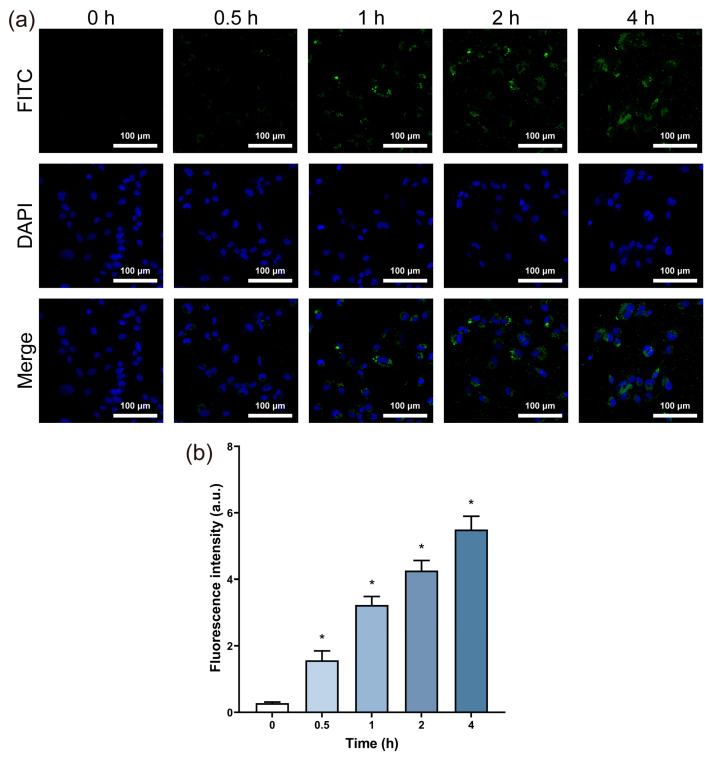
Cell uptake of BPNSs exposed to HUVECs. The HUVEC cells were exposed to the cell culture medium containing 10 μg/mL FITC-labeled BPNSs for different time periods of 0.5, 1, 2 and 4 h. (**a**) Fluorescence images of HUVECs after exposure to 10 μg/mL FITC-labeled BPNSs for 0, 0.5, 1, 2 and 4 h. (**b**) FITC fluorescence intensity was quantitatively analyzed in the cells. * denotes statistical significance with a *p*-value of less than 0.05 compared to the control.

**Figure 3 jfb-14-00284-f003:**
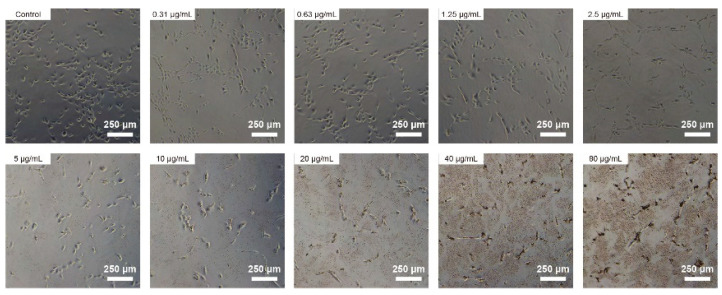
Morphological changes of HUVECs exposed with BPNSs. Morphological images of the HUVEC cells treated with varying concentrations of BPNS (ranging from 0 to 80 μg/mL) in cell culture medium for 24 h.

**Figure 4 jfb-14-00284-f004:**
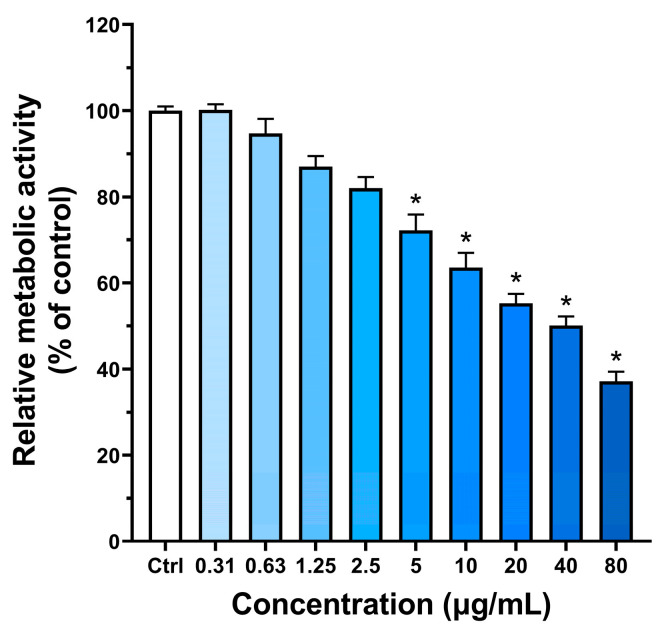
Relative metabolic activity of HUVECs exposed with BPNSs. The relative metabolic activity of HUVECs after treatment with different concentrations of BPNSs (0–80 μg/mL) for 24 h was calculated by the CCK-8 method. * denotes statistical significance with a *p*-value of less than 0.05 compared to the control.

**Figure 5 jfb-14-00284-f005:**
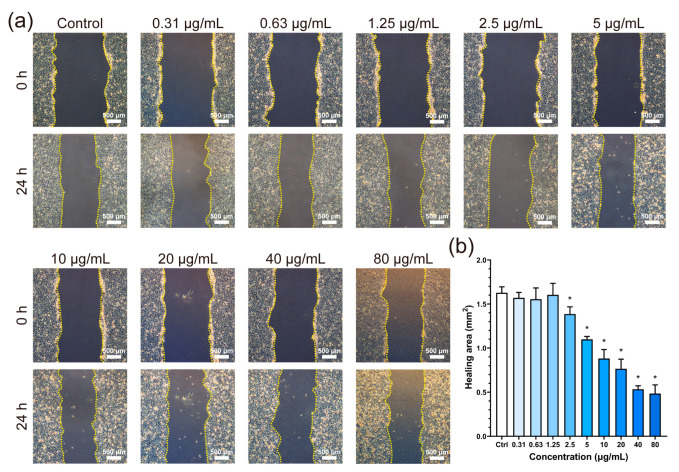
Evaluation of the impact of BPNSs on cellular migration capability. (**a**) Micrographs displaying the cell scratch zone of HUVECs treated with various concentrations of BPNSs for 0 h and 24 h. (**b**) Quantitative analysis of the healing area in the cell scratch region over 24 h. * denotes statistical significance with a *p*-value of less than 0.05 compared to the control.

**Figure 6 jfb-14-00284-f006:**
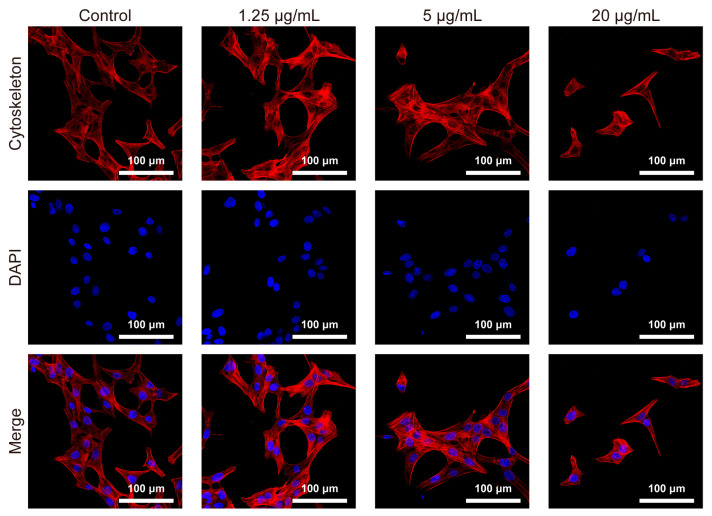
Impact of BPNSs exposure on HUVECs cytoskeleton morphology. Fluorescence microscopy images of phalloidin-stained cytoskeleton and DAPI-stained nuclei of HUVECs incubated with different concentrations (1.25, 5 and 20 μg/mL) of BPNSs for 24 h.

**Figure 7 jfb-14-00284-f007:**
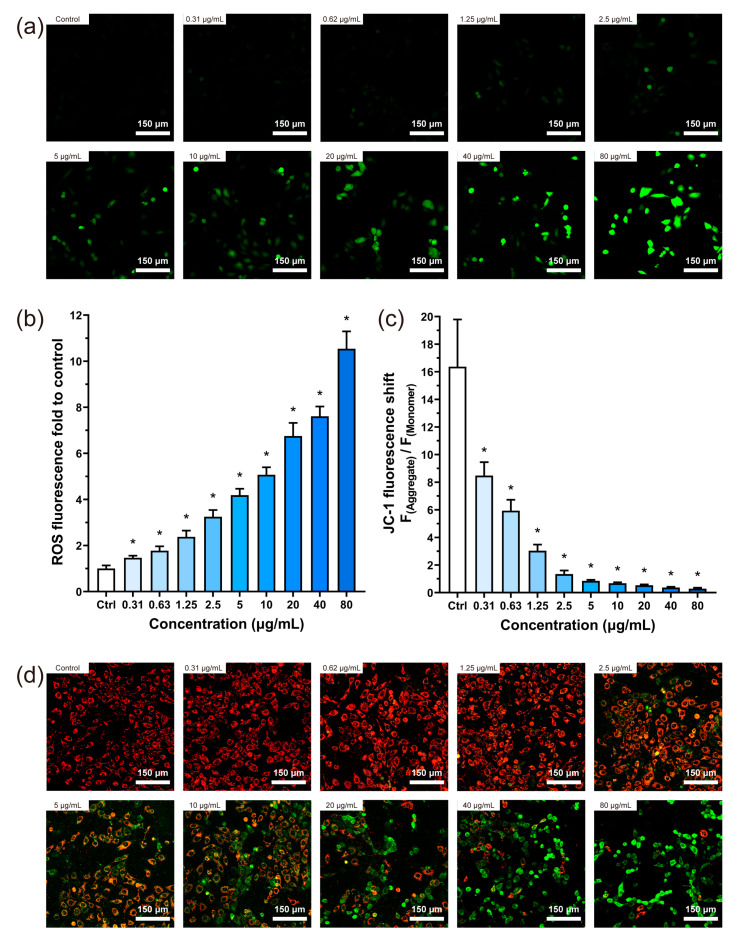
Effects of BPNSs on intracellular ROS levels and mitochondrial membrane potential in HUVECs. (**a**) Fluorescence images were produced by the reaction of ROS with fluorescent probes in the HUVEC cells following 24 h incubation with varying concentrations of BPNSs. (**b**) Quantitative assessment of the relative intracellular ROS levels in HUVECs after 24 h exposure to BPNSs. The relative ROS levels were computed as the fold change in fluorescence intensity compared to the control group. (**c**) Quantitative assessment of the mitochondrial membrane potential changes in HUVECs after 24 h exposure to BPNSs. The relative mitochondrial membrane potential changes were computed as the ratio of intracellular JC-1 red and green fluorescence. (**d**) Following incubation of the HUVEC cells with varying concentrations of BPNSs for 24 h, fluorescent images of mitochondrial membrane potential were obtained using the JC-1 fluorescent probe. * denotes statistical significance with a *p*-value of less than 0.05 compared to the control.

**Figure 8 jfb-14-00284-f008:**
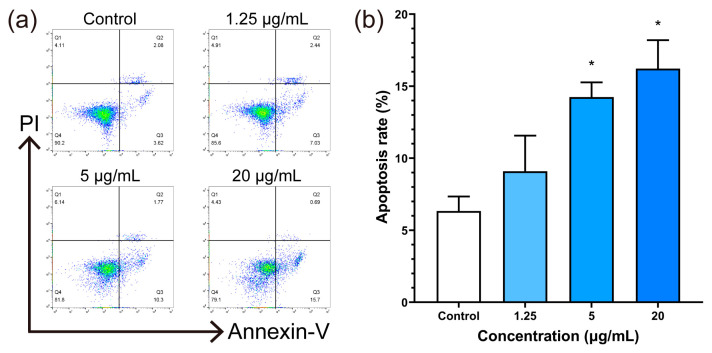
Apoptotic effects of BPNS exposure on HUVECs. (**a**) Flow cytometry analysis of HUVEC apoptosis after exposure to different concentrations of BPNSs for 24 h, and (**b**) corresponding quantitative analysis. * denotes statistical significance with a *p*-value of less than 0.05 compared to the control.

**Figure 9 jfb-14-00284-f009:**
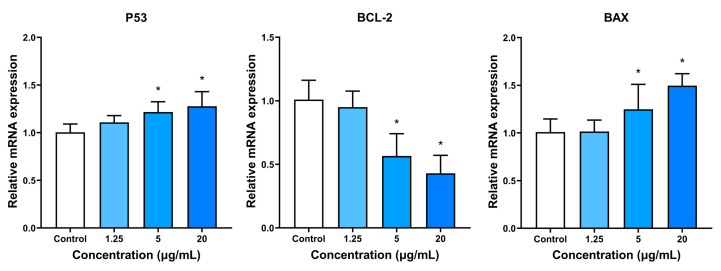
Assessment of the impact of BPNS exposure on apoptosis-related gene expression in HUVECs. RT-PCR analysis was conducted to evaluate the expression levels of P53, BCL-2, and BAX in HUVECs after incubation with different concentrations (1.25, 5 and 20 μg/mL) of BPNSs for 24 h. * denotes statistical significance with a *p*-value of less than 0.05 compared to the control.

**Figure 10 jfb-14-00284-f010:**
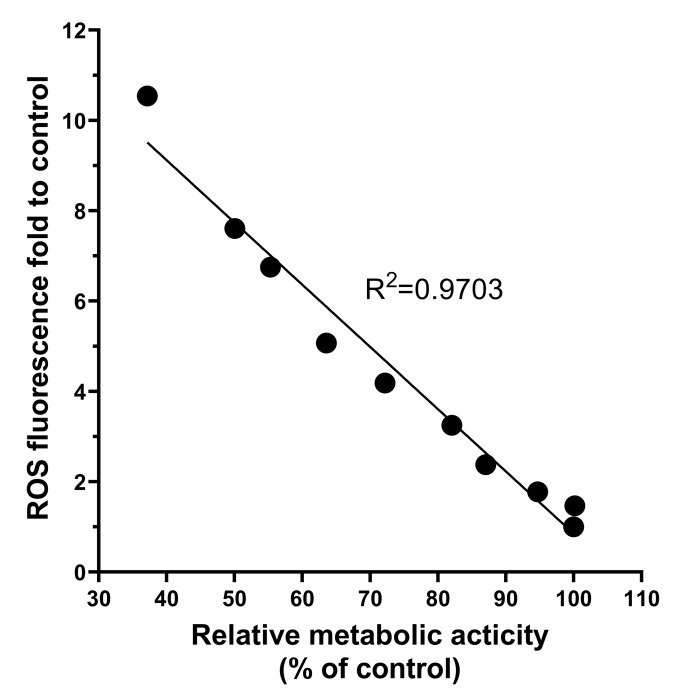
Correlation analysis of BPNSs on the HUVEC cell survival rate and relative intracellular ROS level. The data on relative metabolic activity and ROS levels of HUVECs were used to conduct correlation analysis after 24 h incubation with BPNSs.

**Figure 11 jfb-14-00284-f011:**
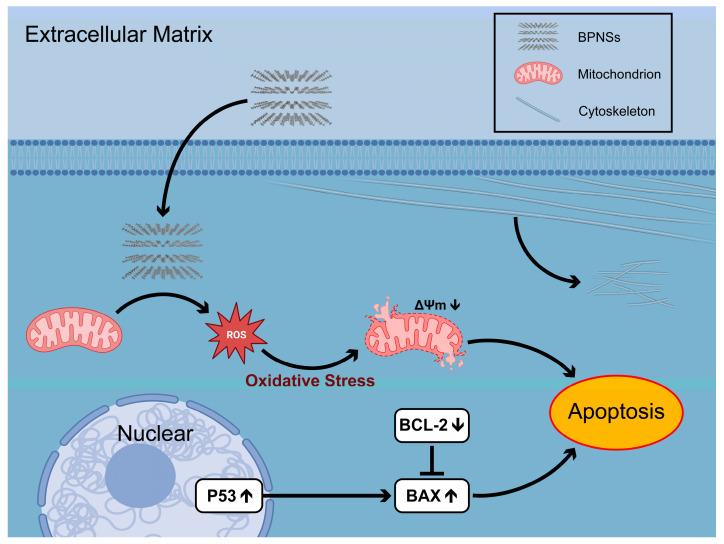
Schematic mechanisms of BPNSs-induced cytotoxicity of HUVECs. The internalization of BPNSs by HUVECs had an impact on the cytoskeletal structure, which was manifested in the form of alterations in cellular morphology and a subsequent decrease in the migration ability. After internalization, BPNSs induced excessive ROS and further led to the dysfunction of mitochondria. These sequential cellular responses promote the progression of cell apoptosis, which further affects cell activity and ultimately leads to reduced cell survival. Abbreviations: BPNSs: Black phosphorus nanosheets; ROS: Reactive oxygen species.

**Table 1 jfb-14-00284-t001:** Primer sequences employed for real-time polymerase chain reaction (PCR) in this investigation.

Gene	Forward Primer Sequence (5′-3′)	Reverse Primer Sequence (3′-5′)
P53	TGTGACTTGCACGTACTCCC	ACCATCGCTATCTGAGCAGC
BCL-2	GAACTGGGGGAGGATTGTGG	CATCCCAGCCTCCGTTATCC
BAX	GAGCAGCCCAGAGGCG	GGAAAAAGACCTCTCGGGGG
18s rRNA	CAGCCACCCGAGATTGAGCA	TAGTAGCGACGGGCGGTGTG

## Data Availability

Access to the data could be requested by contacting the corresponding author.

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
