# Peer review of "Cytotoxicity Induced by Black Phosphorus Nanosheets in Vascular Endothelial Cells via Oxidative Stress and Apoptosis Activation"

_jfb, 2023, doi:10.3390/jfb14050284_

Round 1

Reviewer 1 Report

The paper described the effect of the exosure of HUVEC to Black Phosphorus nanosheets.

Out of the cell model, the paper shows no originality, apoptosis was already demonstrated in other models.

The methodology is classic, and the experimental design made to support the demonstration.

Major comments:

The decision to use a data not supported by the statistical analysis strongly deserve the paper.

Line 375 - 387 : the sentence is not true, there is no evidence supporting an upregulation  of p53 in this experiments set. Only BAX and BCL-2 were regulated after BPNCs exposure. The authors should rewrite this argumentation, without over interpretating the datas. There is no experimental evidence that down regulation of BCL-2 is due to some up-regulation of p53.

Figure 11 has to be draw again, without p53.

Line 409-410 : remove any quotation to p53 being involved, no data in this experimental set support this.

The fact that the number of samples per study group are not provided strongly deserve the paper. Does the figure refer to one experiment , or a representative one from a large number ? a biological triplicate ? a technical triplicate ?

The fact that no good review about apoptosis is quoted to support the discussion strongly deserve thepaper. It suggests the authors did not know the basics, and just take some ideas from various paper in the nanomaterial field ! I suggest the authors refer to some more general review about apoptosis instead of experimental papers, for instance : doi: 10.1111/febs.14186.

For PCR datas, the addition of at least one housekeeping gene would increase the meaning of the change in the expression of BCL-2 and BAX genes.

Minor comments:

In the abstract

"The inhibition of cell proliferation was determined by the concentration of BPNSs" ? This sentence is a nonsense.

The legends of quite all figures are way too short. They must allow the reader to understand the figures without reading the text.

In the text

Line 18-19 : lack of capital at the beginning of the sentence "when the concentration …"

Line 226-227: how did the authors count the cells? No data, no figure support the sentence in these lines. And no data nor figure is quoted.

Lines 320 and 321: I would be delighted if the authors could explain how they cultivate BPNS. Like any cells? remove co-culture!

Figures:

Figure 7

a and b: legend indicate ROS production, not DCF nor DCFH

c and d: legend present JC-1 data, without signification sense

At least the authors should be consistent in the legend!

Figure 10 legend should quote the ROS data origin, as well as the cell viability data origin.

In the Discussion

Discussion referred to the results, but should quote the figures;

Ref 43 is too specific, and for the discussion's sake a good review would be more appropriate.

Reviewer 2 Report

The study is well organized and presented well. However, some of my comments need to be answered before considered for publication.

Abstract: These findings provide significant information for the potential applications of BP - seems to be incomplete, I suggest to "application of BP in tissue engineering"?

The AFM image is not clear, and perhaps the AFM is nowhere matching with the TEM results. The TEM image is also unconvincing, as only single BNPs were demonstrated.  

Does the cell migration seem the (white color) dead cells were observed at all the treated wells?

(2,7-dichlorodihydrofluorescein diacetate, DCFH-DA)? both are the same?

Why in the result section it is written DCFH to DCF? instead of using DCFH-DA to ? 

The cell migration at 2.5 concentration is significant to highlight as mentioned in the conclusion, suppressed cell migration.

Reviewer 3 Report

As a bioactive material for tissue engineering, black phosphorus (BP), a developing two-dimensional material with distinctive optically, thermoelectric, and mechanical characteristics, has been presented. Its harmful effects on physiological mechanisms, however, are unknown. Therefore, Dong et al aimed to investigate BP's cytotoxicity to endothelial cells in the vascular system. They found BPNSs could influence the expression of apoptosis-related genes such as the P53 and BCL-2 families, leading to HUVEC apoptosis. As a result, BPNS concentrations of more than 2.5 g/mL could impact the viability and activity of HUVECs. 

I found, that the topic is original and relevant in the field.

The conclusion appears to me to be consistent with the data and arguments presented.

The manuscript is interesting; nonetheless, it may be enhanced and improved by addressing the following points -

The captions of Figures 3 and 6 are not clear.

Figure 7 should be provided in high resolution.

The p53 is the well-known controller for cell deaths such as apoptosis, autophagy, ferroptosis....etc.The authors should discuss more p53 in the Discussion and cite these studies PMID: 32987354, PMID: 9671594, PMID: 29800655...etc

Figure 11 looks good but is hard to understand.
